# Tumor-Associated Macrophage Status in Cancer Treatment

**DOI:** 10.3390/cancers12071987

**Published:** 2020-07-21

**Authors:** Anna Maria Malfitano, Simona Pisanti, Fabiana Napolitano, Sarah Di Somma, Rosanna Martinelli, Giuseppe Portella

**Affiliations:** 1Department of Translational Medical Sciences, University of Naples Federico II, 80131 Naples, Italy; fabiananapolitano@libero.it (F.N.); sarah_ds@hotmail.it (S.D.S.); 2Department of Medicine, Surgery and Dentistry “Scuola Medica Salernitana”, University of Salerno, Via Salvador Allende, Baronissi, 84081 Salerno, Italy; spisanti@unisa.it (S.P.); rmartinelli@unisa.it (R.M.)

**Keywords:** tumor-associated macrophages, chemotherapy, radiotherapy, immune-checkpoint blocking therapy, oncolytic virus, virotherapy

## Abstract

Tumor-associated macrophages (TAMs) represent the most abundant innate immune cells in tumors. TAMs, exhibiting anti-inflammatory phenotype, are key players in cancer progression, metastasis and resistance to therapy. A high TAM infiltration is generally associated with poor prognosis, but macrophages are highly plastic cells that can adopt either proinflammatory/antitumor or anti-inflammatory/protumor features in response to tumor microenvironment stimuli. In the context of cancer therapy, many anticancer therapeutics, apart from their direct effect on tumor cells, display different effects on TAM activation status and density. In this review, we aim to evaluate the indirect effects of anticancer therapies in the modulation of TAM phenotypes and pro/antitumor activity.

## 1. Introduction

TAMs are key players in cancers influencing progression, metastasis and tumor recurrence. TAMs originate mainly from circulating precursor monocytes, however, resident macrophages can later develop in a tumor [1,2]. Specific signals by chemokines, i.e., CCL2 and colony stimulating factors (CSF)-1, cytokines and components of complement recall monocytes to tumor sites [3]. 

In solid tumors, macrophages represent the main immune population constituting up to 50% of the tumor mass. Macrophage plasticity allows these innate immune cells to adopt their well-known M1–M2 polarization axis. TAMs usually display a protumor/anti-inflammatory phenotype associated with the M2 profile, whereas the antitumor/proinflammatory function is associated with the M1 phenotype. TAMs and M2 macrophages sustain tumor growth and progression, inhibiting immune-stimulatory signals. TAM infiltration in tumor has been associated with poor prognosis [3]. Numerous investigations suggest TAMs as an interesting therapeutic target since strategies aiming to deplete TAM, inhibit their recruitment and influence their polarization status might be employed to promote their antitumor effects. Additionally, TAMs are mainly responsible for resistance to well-known antitumor treatments like chemotherapy and radiotherapy, indeed they limit the efficacy of immunotherapy, i.e., anti-PD1 treatment [3,4,5]. However, depending on the tumor type and treatment adopted it might be possible to identify novel approaches, i.e., combinatory therapies also taking advantage of more recent treatments as those based on the use of oncolytic viruses (OV) [6,7,8,9]. In this review, we will highlight TAMs as therapeutic target and their modulation by anticancer therapies.

## 2. Classically Activated vs. Alternatively Activated Macrophages

### 2.1. Proinflammatory/Antitumor M1 TAMs

Classically activated macrophages (M1) are crucial for host defense and cancer cell killing. M1 TAMs are stimulated by toll-like receptor (TLR) ligands, microbial substrates such as lipopolysaccharide (LPS) and Th1 cytokines. Once activated, M1 macrophages secrete proinflammatory cytokines and express markers typically used to identify the M1 phenotype [10] (Figure 1).

### 2.2. Anti-Inflammatory/Protumor M2 TAMs

The alternatively activated macrophages (M2) are involved in the resolution of inflammation and suppress the immunity against parasites and tumor cells, thus enabling the tumor microenvironment to promote cancer progression and metastasis. M2 TAMs are activated by Th2 cytokines, transforming growth factor β (TGFβ), chemokines and prostaglandin E2 (PGE2). M2 TAMs secrete anti-inflammatory cytokines, TGFβ and express M2 phenotype markers (Figure 1). Additionally, the M2 profile is characterized by upregulation of genes such as arginase 1 (Arg1), resistin-like molecule α (FIZZ1), macrophage mannose receptor (MMR) 1, (Mrc1) and chitinase-like protein Ym1 [10].

## 3. Role of TAMs in the Tumor Microenvironment

The dynamic and heterogeneous interactions between cancer cells and tumor microenvironment determine a range of opposite functions of macrophages even within the same type of tumor. Tumor and immune cells secrete growth factors, cytokines and metabolites that promote TAM protumor polarization. In particular, mediators like CSF-1, CCL2 and vascular endothelial growth factor (VEGF) promote TAM accumulation in the tumor microenvironment [11,12,13]. The protumor activation status of macrophages includes secretion of growth factors, promotion of angiogenesis, release of proteases and molecules for extracellular matrix remodeling, production of immune-suppressive molecules. Among molecules affecting cancer cell proliferation, TAMs express epidermal growth factor (EGF), members of the TGFβ and fibroblast growth factors (FGF) [4,14]. TAMs also promote vessel growth by upregulation and release of proangiogenic factors such as VEGF, tumor necrosis factor (TNFα), FGF, urokinase plasminogen activator (uPA), thymidine phosphorylase (TP), adrenomedullin (ADM) and semoforin 4D (Sema4D) [14,15]. TAMs release proteolytic enzymes such as matrix metalloproteinases (MMPs) and cathepsins able to degrade the extracellular matrix activating and releasing angiogenic factors [3,16,17]. Additionally, TAMs, by the release of proangiogenic chemokines and cytokines such as IL10, TGFβ, CCL3, CCL4, CCL5 and CCL22, mediate the immunosuppressive activity on T cells by repression of genes encoding granzymes, perforins and cytotoxins [18,19,20]. Indeed, secreting proinflammatory cytokines, such as IL6 and TNFα, TAMs, contribute to the “dormant inflammation” determining immunosuppression in the tumor microenvironment.

## 4. Exploiting TAMs as a Therapeutic Target

TAM subpopulations are implicated in all the steps of tumor growth and progression, including epithelial-mesenchymal transition, invasion, angiogenic switch, intravasation, extravasation, priming of the metastatic niche [16,21]. The complex interplay between tumor cells and TAMs orchestrate tumor progression and metastasis formation through the secretion by both cell types of cytokines, chemokines and growth factors sustaining tumor growth. Among the rational approaches to counteract tumor progression, the study of TAM targeting and reprogramming is aimed at creating an unfavorable environment for tumor cells thus preventing their exploitation of hosting immune cells. Therefore, TAM targeting may help not only to reduce the growth of the primary tumor, but also to hinder metastases formation. Strategies able to target TAMs and modulate their reprogramming are below reported and showed in Figure 2.

### 4.1. Strategies Aimed at TAM Depletion

The first strategy of TAM targeting is based on TAM depletion. Several molecules tested in preclinical models showed a promising efficacy in clinical trials. 

Trabectidin is an antineoplastic alkylating agent of marine origin, approved for the treatment of advanced soft tissue sarcoma and ovarian cancer relapse and under testing in clinical trials for other solid tumors. Beyond a direct cytotoxic effect on tumor cells, its antitumor activity relies also on the depletion of TAMs (in a range from 30 to 77% as observed in clinical studies) through induction of apoptosis [22]. Structurally and functionally similar to trabectidin, lurbinectedin has shown promising clinical results in the treatment of small cell lung cancer, alone or in combination with checkpoint inhibitors or conventional chemotherapeutics [23,24]. Another class of drugs that showed secondary efficacy in depleting TAMs when used in chemotherapeutic regimens, are bisphosphonates, originally developed as inhibitors of osteoclastic bone resorption, now used in both hematological and solid tumors to contrast bone metastases and skeletal negative events [25]. Liposomal formulations of clodronate have been reported to target macrophage survival and functions reducing bone and visceral metastases in breast cancer and others [16,26]. The antitumor properties of zoledronate rely also on the impairment of TAM polarization, neoangiogenesis and metastatization. Lipid-coated calcium zoledronate nanoparticles, beyond inducing TAM apoptosis, were able to inhibit angiogenesis and improve antitumor immune response [27].

Another interesting target is the CSF1 receptor (CSF1R) that is expressed on circulating monocytes and tissue macrophages and controls their survival, proliferation, differentiation and chemotaxis. CSF1/CSF1R signaling axis is overexpressed in several tumors and associated to a worse prognosis [28]. Monoclonal antibodies or small tyrosine kinase inhibitors to block CSF1R signaling showed efficacy in preclinical models, successfully depleting TAM and enhancing the percentage of intratumor T cells. Three CSF1R-blocking mAb, RG7155 (emactuzumab), IMC-CS4 and FPA008 are in clinical evaluation, alone or in combination therapy, in several solid tumors [29,30]. In mice models of breast cancer and melanoma, CSF1R-blocking antibodies were not able to arrest the growth of the primary tumors but significantly reduced the development of metastases [31]. Only a partial response was observed for emactuzumab in combination with paclitaxel for the treatment of advanced solid tumors, even if M2 TAM depletion and repolarization toward M1 was obtained [32]. Small molecule tyrosine kinase inhibitors PLX3397 (pexidartinib) and BLZ945 provided more encouraging results both in preclinical models and in the clinic. Pexidartinib has been recently authorized by the FDA for the chemotherapy of inoperable or relapsing diffuse-type tenosynovial giant-cell tumor, a rare aggressive tumor of the connective tissue caused by a translocation associated to CSF1/CSF1R overexpression [33]. The clinical use of both antibodies and small molecules was associated to the occurrence of severe adverse events, such as fatigue, asthenia, anemia, nausea, facial and periorbital oedema, lupus erythematosus and hepatotoxicity [34]. Such side effects are likely due to unwanted disruption of perivascular macrophages in non-target tissues and organs. Indeed, none of above reported approaches leads to the selective elimination of protumorigenic M2-like macrophages without affecting the protective populations that maintain their natural antitumoral function or perform specific protective actions in various body’s districts. More selective strategies are thus being developed, like the promising M2-macrophage-targeting peptide (M2pep) able to preferentially identify and kill M2 TAM saving up the antitumor M1 subpopulation [35]. Another possibility is to target the scavenging receptor CD163, an M2 and TAM marker, depleting selectively such subpopulation reported to inhibit melanoma progression in mice through the promotion of M1 and T cell-mediated antitumor immunity [36]. 

### 4.2. Strategies Aimed at Inhibiting TAM Recruitment

Several primary tumors showed high expression of both CCL2 and its receptor CCR2 that were associated to TAM recruitment [37]. Preclinical studies with CCL2 blocking antibodies or small molecule inhibitors supported the notion of their key role in the intratumor accumulation of TAMs, through signaling pathways that are strictly dependent on the tumor type [38]. The inhibitors were efficacious alone or in combination with other chemotherapeutics in various tumor models, even if drug removal elicited the recruitment of new macrophages and the rapid development of metastases in breast cancer [39]. Despite their promising efficacy in preclinical studies, clinical trials with CCL2/CCR2 inhibitors were partially disappointing. The anti-CCL2 monoclonal antibody carlumab failed to inhibit tumor growth in early stage clinical trials in prostate cancer, since CCL2 levels increased just one week after the treatment through the induction of compensatory mechanisms [40]. Better results were obtained in combination regimens with the CCR2 inhibitor PF-04136309 and FOLFIRINOX chemotherapy (leucovorin, fluorouracil, irinotecan, oxaliplatin) in advanced pancreatic cancer, where reduced numbers of both circulating and intratumoral CCR2+ monocytes were measured. The treatment was well tolerated, and an objective tumor response was reported in half of the patients. In the FOLFIRINOX group none objective response was observed [41]. On the contrary, the combination of PF-04136309 with paclitaxel and gemcitabine in metastatic pancreatic cancer displayed safety issues and had no therapeutic advantage with respect to the standard treatment [42]. The partial response to CCL2/CCR2 targeting strategies highlights that other chemokines, cytokines and growth factors contribute to the process, or may compensate the absence of CCL2/CCR2, with differences depending on the stage and the type of tumor, that hence need to be carefully evaluated for the development of effective strategies.

Other monoclonal antibodies and small molecule inhibitors targeting chemokine signaling pathways are already available for the treatment of autoimmune pathologies, like CCR5 antagonists. CCR5 is overexpressed in various solid cancers and in lymphoma, being also related to a worse prognosis. CCR5 antagonists, like maraviroc and vicriviroc, used as antiretroviral drugs in HIV combination regimens (since CCR5 acts as a coreceptor for HIV-1 entry into CD4+ lymphocytes) have been repurposed in cancer [43]. Both maraviroc and vicriviroc were efficacious in preclinical models of breast and colon cancer through the induction of antitumor immunity [44]. In colorectal cancer patients, refractory to other therapeutic regimens, maraviroc showed a good safety profile [45]. CXCL12/CXCR4 and CCL20/CCR6 are alternative signaling axes that could be successfully targeted in cancer to prevent TAM accumulation. The CXCR4 antagonist AMD3100 (plerixafor) has been reported to restrain tumor growth also through the inhibition of immunosuppressive mechanisms [46]. Several clinical trials are ongoing to test plerixafor in solid tumors, in order to assess its safety profile alone or in combinatorial regimens. 

Development of dual-antagonists targeting simultaneously two different chemokine receptors is a strategy that could lead to more effective TAM targeting. The dual CCR2/CCR5 antagonist BMS-813160 is actually under clinical investigation in combination with nivolumab or standard chemotherapy in pancreatic, lung, renal and hepatocellular cancers.

Chemokine-targeting agents need to be accurately selected in order to not affect the recruitment of other immune cells like natural killer (NK) and T cells that are fundamental for the efficacy of immunotherapeutic approaches. Of note, recent studies have reported that cytotoxic cells mainly use CXCR3 for intratumor homing [38]. Thus, targeting chemokine receptors other than CXCR3 may be effective in inhibiting TAM recruitment alone.

Understanding the interplay among different factors in the microenvironment and their correlation to TAM homing in different tumors, will help the development of efficient targeted therapies aimed at inhibiting TAM recruitment or retention both in the primary tumor and at metastatic sites avoiding the priming of compensatory mechanisms.

### 4.3. Strategies to Influence TAM Polarization

The above-reported strategies indiscriminately target all macrophages in the tumor and other districts, inducing undesired side effects and long-term toxicities. Reprogramming TAMs has the advantage to address them specifically toward the antitumor phenotype. Modulation of cytokines, chemokines, growth factors and their receptors can direct TAM functional program. Many of these factors are particularly abundant in hypoxic conditions, a frequent characteristic of solid tumors. The direct or indirect induction of proinflammatory mediators, like IFNγ, IL4, IL6, IL13, VEGF, GM-CSF, CSF-1, Ang-2, CCL2 and other chemokines in the tumor milieu, has an antitumor effect linked to M1 polarization of TAM. An oncolytic virus encoding the proinflammatory cytokine IL12 has been reported to efficiently kill glioblastoma tumors also through the restoration of the antitumor immune response [47]. 

Among the numerous approaches in preclinical models, including small molecules, antibodies and RNA, some are giving promising results also in clinical testing, alone or in combination with classical chemotherapeutic drugs or immune checkpoint inhibitors. 

An interesting strategy aims to enhance TAM phagocytic properties blocking the CD47/SIRPα ‘don’t eat’ signal by target cancer cells that express CD47 at high levels as an acquired mechanism of resistance to clearing by phagocytosis [48]. Pharmacological inhibition of CD47 signaling by the antibodies Hu5F9-G4 and CC-90002 or the engineered fusion protein SIRPα-Fc (TTI-621) was effective in enhancing tumor cell phagocytosis. The combinatorial treatment with monoclonal antibodies like rituximab further improved the antitumor response through phagocytosis induction in lymphoma [49]. Another way to reprogram TAMs toward M1 phenotype and augment phagocytosis is through the anti-CD40 mAb selicrelumab that is now under testing in Phase I trials in several solid tumors alone or in combination with standard chemotherapy or immunotherapy. Moreover, its combination with emactuzumab has been reported to be particularly effective in the repolarization of TAMs without undesired depletion of all macrophages [50].

The activation of TLR by bacterial products or viral nucleic acids is able to induce the immune response, addressing macrophages toward the M1 phenotype. Encouraging results were obtained in mice tumor models of melanoma, breast cancer and others, with TLR3, TLR7, TLR8 and TLR9 ligands [16]. The TLR7 agonist imiquimod, approved for the topical treatment of squamous and basal cell carcinoma, was reported to be safe also in the treatment of melanoma and skin lesions of metastatic breast cancer, enhancing accumulation of immune cells in the lesions [51]. Poly-I:C stimulation of TLR3 efficiently repolarized TAMs in melanoma and now is under evaluation as a cancer vaccine to improve the antitumor immune response in advanced cancers [52]. The scavenger receptor MARCO (Macrophage Receptor with Collagenous Structure), of which engagement is fundamental for pattern recognition receptor responses, is highly expressed by TAMs in patients with melanoma and breast cancer and found to be related to a worse prognosis. An antibody specifically directed to MARCO receptor has been reported to redirect TAMs toward the inflammatory phenotype, increasing the antitumor immune response also in combination with anti-CTLA-4 immune checkpoint inhibitor [53].

Several small molecule inhibitors exerted their antitumor action activating the immune system through the reprogramming of macrophages. The inhibition of PI3Kγ blocked tumor growth through the induction of a proinflammatory response by M1 TAMs that recruited CTLs into the tumor [54]. Mice treated with PI3Kγ inhibitor in combination with anti-PD1 checkpoint inhibitor had improved response and overall survival [55].

Another promising strategy is TAM repolarization at the epigenetic or genetic level. The histone deacetylase inhibitor TMP195 is able to address TAMs toward M1 profile, modifying their epigenetic signature and inducing their accumulation and phagocytic activity in the microenvironment of breast tumors [56]. miRNA modulation through the genetic deletion in macrophages of DICER, an enzyme fundamental for miRNA synthesis, was shown to inhibit tumor growth in preclinical models of breast, colon and lung tumors, reprogramming TAMs to express IFNγ and activate STAT1 proinflammatory signaling [57]. 

Undoubtedly, all approaches targeting TAMs need to be based on a solid mechanistic knowledge of TAM plasticity and ability to switch from a phenotype to another in different phases of tumor growth and progression and in response to therapy.

### 4.4. Targeting TAM Receptors (TAMR)

An interesting druggable target is represented by the so-called TAMR, a family of three known tyrosine kinase receptors, Tyro3, Axl and MerTK, whose initials form the acronym TAM (unintentionally identical to TAM cells’ acronym), expressed by several cell types including tumor cells and immune cells. These receptors display multiple roles in cell fate, proliferation, migration and in modulating processes like tissue homeostasis and inflammation [58]. 

The best-known ligands for TAMR are Gas6 and Protein S that function as adaptors connecting phosphatidylserine proapoptotic signal on target cells with TAMR, thus favoring phagocytosis by macrophages. The activated downstream signaling pathways converge on the PI3K/Akt axis that is involved in TAM activation and polarization towards the M2 phenotype, associated with the promotion of efferocytosis, which limits the inflammatory response, fostering the production of immunosuppressive cytokines and preventing the immune cell activation in response to cancer cell death in solid tumors. For these reasons, blocking TAMR may be a promising anticancer strategy [59]. Several small-molecule inhibitors of TAMR along with antibody–drug conjugates, engineered CAR-T cells and Axl proteins fused with Fc, have been developed as anticancer agents in different preclinical models and some are already in clinical trials. These drugs have been originally designed to target cancer cells expressing TAMR that are upregulated, as well as their partner protein Gas6, in several tumors, like leukemia, melanoma and glioblastoma [60]. Axl expression levels correlated with worse prognosis in glioblastoma and pancreatic cancer, where Axl is overactivated in the 70% of cases and is responsible of resistance to chemotherapy [61,62,63]. Therefore, the action of these molecules on TAM recruitment in animal models and their evaluation as a secondary endpoint in clinical studies is worth of investigation. MerTK is the most abundant receptor of the family on macrophages and microglia, whereas Axl is prevalently expressed by dendritic cells (DCs) [58]. MerTK blocking with a specific antibody leads to inhibition of TAM-mediated efferocytosis of apoptotic tumor cells that activate an inflammatory type I IFN response through the induction of the stimulator of interferon genes (STING)-controlled innate immune pathway, with subsequent CD8+ activation and a synergistic effect with anti-PD1/PDL1 checkpoint inhibitors [64]. The blood–brain barrier permeable small molecule UNC2025 has been observed to inhibit glioblastoma growth in combination with radiotherapy [60]. The Axl inhibitor BGB324 showed promising efficacy in a preclinical model of glioblastoma in combination with nivolumab anti-PD1 antibody [62]. A pan-TAM inhibitor, RXDX-106, able to block all the receptors of the family, was effective in inhibiting tumor progression in several models, boosting antitumor immunity [65]. Strategies to target Gas6 have been developed, by blocking antibodies or drugs like warfarin, and are currently in an early Phase 1 clinical trial in patients with pancreatic cancer. Beyond inhibiting epithelial mesenchymal transition of pancreatic tumor cells, targeting Gas6 promotes the antitumor immune response mediated by NK cells [66].

## 5. TAMs in Cancer Treatment: Chemotherapy, Radiotherapy, ICB Therapy and Virotherapy

The ideal cancer treatment is expected to switch TAMs toward an antitumor phenotype. Here, we discuss the main findings describing the effect of conventional and more recent anticancer therapies on TAM reprogramming.

### 5.1. TAMs and Chemotherapy

TAM behavior during chemotherapeutic treatment is controversial, and their potentiated or reduced effect on chemotherapy depends on the chemotherapeutic agent and type of tumor (Table 1). Three major mechanisms seem to drive the editing of macrophages by chemotherapeutics: (i) polarization, (ii) recruitment and migration and (iii) depletion.

#### 5.1.1. Polarization

Numerous chemotherapeutics prompt the antitumor M1 phenotype switch from the M2 population. In vitro and in vivo, paclitaxel induced M1 macrophage polarization in a TLR4-dependent manner [79]. In addition, paclitaxel enabled macrophages to activate genes encoding inflammatory mediators (see Table 1) and cytokines able to activate NKs, DCs and tumor-specific cytotoxic T-lymphocytes (CTLs) [67].

In mammary tumors 4T1-Neu in mice, docetaxel caused the activation of M1-like TAMs and depletion of M2-like TAMs, enhancing CTL response [69]. Cisplatin treatment of murine peritoneal macrophages cocultured with L929 cells, favored the release of cancer cell-specific cytotoxic factors such as FasL and TNF and facilitated the apoptosis of tumor cells, confirmed by the activation of caspase-8, caspase-3, cytochrome c, Bid, Bax, along with DNA fragmentation and downregulation of Bcl-2 [80]. Indeed, cisplatin induced the secretion of IL1, IL6, IL8 and TNFα by peritoneal macrophages isolated from BALB/c mice [81]. On the other hand, in ovarian and cervical cancer cell lines, cisplatin and carboplatin, by increasing the production of PGE2 and IL6 by cancer cells, skewed monocyte differentiation towards the M2 phenotype leading to chemoresistance [70].

In vivo, in mouse peritoneal macrophages, cyclophosphamide enhanced the production of IL6, and IL12 and decreased anti-inflammatory cytokines like IL10 and TGFβ, thus activating Th1 cells and macrophages [82]. Combinatory approaches including the cotreatment with chemotherapy (cyclophosphamide, doxorubicin, vincristine) and immunotherapy (anti-CD40+ cytosine-phosphate-guanosine-containing oligodeoxynucleotide 1826) prompted the enrichment of an M1 polarized TAM subset. Thus, molecules associated with the M1 phenotype were upregulated whereas M2-associated molecules (see Table 1) were downregulated [76].

Chemotherapeutics mostly are effective in reprogramming TAMs versus the M1 phenotype, however, depending on tumor type and therapy schedule, can promote the M2 phenotype. The interaction between TAMs and chemotherapeutics deserves a careful investigation, addressing patient-to-patient differences, tumor types and type of treatment. Nonetheless, in the M1 phenotype prevalence, it must be taken into account whether the proinflammatory effect of TAMs can promote an antitumor response or favor a low-grade inflammation. Further investigations are required to address open questions.

#### 5.1.2. Recruitment and Migration

Chemotherapeutics induce the recall of monocytes to tumor site as consequence of chemotherapy’s damaging effects. TAMs initiating the regenerative program, contribute to support cancer cell proliferation [83]. Among these agents, cyclophosphamide was shown to recruit DCs, macrophages and NKs in mouse models and in cancer patients [71,72]. Combinatory approaches using chemotherapeutics and other agents have been studied [68,77,78,84]. In HER2/Neu-driven mammary carcinoma, doxorubicin in combination with lapatinib, favored the infiltration of immature macrophages and the reduction of mature TAMs at the tumor site [74]. In addition, in a mouse model of breast cancer metastasis, doxorubicin increased the recruitment of monocytes; in particular, CD206+ macrophages favored an increased vascular leakage that prompted a better doxorubicin response, whereas CCR2-dependent monocyte recruitment was associated with tumor relapse [73]. In vivo, in murine breast tumor, abraxane (albumin + paclitaxel) evoked high infiltration of CD45+CD169+ macrophages, an increased F4/80+ macrophage population was detected in MDA-MB-435 tumors but not in the correspondent paclitaxel resistant tumors [68]. The treatment with gemcitabine of patients with pancreatic cancer induced increase of CD14+ monocytes, CD123+ plasmacytoid DCs and CD11c+ myeloid DCs [75]. The conventional chemotherapy (oxaliplatin, docetaxel, irinotecan with folinic acid and 5-fluorouracile, 5-FU) combined with an anti-CCL2 to block monocyte migration provided an increased antitumor response in pancreatic and prostate cancer [77,78]. Overall, the pro/antitumor role of infiltrating monocytes/macrophages at tumor sites depends on the type of chemotherapeutic and the local tumor microenvironment.

#### 5.1.3. Depletion

Chemotherapy is known to lead to monocytopenia [85,86]. The effect of docetaxel in 4T1-Neu mammary tumors involves the depletion of the M2 TAMs and activation of M1 monocytes and myeloid-derived suppressor cells that can promote antitumor cytotoxic T cell response [69]. In murine tumors, trabectedin exerted cytotoxic effects on monocytes/macrophage in the bloodstream, bone marrow and spleen (Table 1). Patients with soft tissue sarcoma treated with trabectedin showed a strong reduction of TAM density [22].

### 5.2. TAMs and Radiotherapy

Radiotherapy can affect TAM phenotype and polarization according to the dosage used (Table 2). Radiotherapy induces the production of antioxidant molecules conferring resistance to macrophages that, in humans, are considered the most radio-resistant cells [87]. Radiotherapy prompts tumor regression, however, the recruitment of macrophages and myeloid cells at tumor site might lead to tumor recurrence [88,89,90]. Liposamol clodronate used to deplete macrophages before ionizing radiation renders efficient the antitumor effects of radiotherapy and highlights the protumor role of TAMs [91]. Macrophage infiltration and accumulation by radiotherapy is mainly mediated by CCL2 and CSF1 [90,92]. 

#### 5.2.1. Low Doses

Low doses (below 1Gy) evoke low toxicity and favor the M2 phenotype of TAMs. In Raw264.7 macrophages, low irradiation doses decreased iNOS level and NO production, repolarizing M1 macrophages towards the M2 phenotype [93]. In addition, in the same cells stimulated with LPS, irradiation decreased p38 phosphorylation and TNFα secretion [94]. An anti-inflammatory phenotype was also observed in the LPS-stimulated human THP-1 macrophage cell line with a decrease of IL1β [95,96]. Doses under 2Gy do not affect macrophages’ viability and phagocytic ability but induce an anti-inflammatory cytokine milieu, i.e., increased TGFβ secretion favoring an M2 phenotype [97]. In whole-body-irradiated mice [101], different effects were observed depending on mouse strain (see Table 2). 

#### 5.2.2. Intermediate Doses

Intermediate doses between 1 to 10 Gy induce a shift of unpolarized macrophages toward the M1 phenotype. These macrophages show upregulation of proinflammatory markers and downregulation of M2 anti-inflammatory markers [98] (Table 2). Indeed, the observed M1 phenotype correlated with increased phagocytosis, and irradiation did not affect in macrophage-tumor cell coculture the promotion of invasiveness and angiogenesis [98]. A proinflammatory profile was also associated with increased mRNA levels of IFNγ, TNFα, IL23, IL6 and higher protein levels of IL8 and IL1β elicited by the transcriptional expression of IRF5 and mediated by ataxia telangiectasia mutated (ATM) [102]. Moderate doses can also potentiate an already acquired M1 profile, whereas, did not affect a prompted M2 profile, (Table 2) [98]. Additionally, in coculture experiments using human unpolarized macrophages with radiosensitive and/or radio-resistant colon cancer cells, the irradiation induced different effects. The irradiation in the coculture with radiosensitive cells prompted the decrease of proinflammatory markers (CXCL8, CCR7, IL1β) while anti-inflammatory marker expression was not affected. In the coculture with radio-resistant cells, proinflammatory (CD80, CCR7) and anti-inflammatory (CCL18, IL10) markers were upregulated, these findings suggest that the interaction with different cancer cell types can affect the macrophage phenotype [103]. 

In vivo experiments performed in whole-body-irradiated mice, intermediate doses increased the M1 markers and reduced M2 markers, and in contrast, local irradiation did not change M1 and M2 marker expression. The use of local irradiation determined a reduction of T cell infiltration, whereas local irradiation associated with CTL transfer enhanced M1 and reduced M2 markers (Table 2). Indeed, T cell infiltration was proinflammatory-macrophage-dependent. These data suggest a role of CD8+ T cells in macrophage reprogramming [89]. A shift toward M1 phenotype was observed by systemic irradiation (2Gy/week for two weeks) in mice (Table 2). In addition, whole-body irradiation enhanced of iNOS levels and NO production, inducing tumor vasculature normalization [88]. It was proposed that whole-body irradiation differently from local irradiation favors the infiltration of fresh reprogrammed macrophages from different lymphoid organs 

#### 5.2.3. High Doses 

Doses higher than 10 Gy switch macrophages toward the M2 phenotype. In vitro, irradiation (20 Gy) of Raw264.7 macrophages induced the M2-like phenotype. This effect was confirmed in mice with Panc02 xenograft (Table 2) [99]. High irradiation increased M2-like TAMs also in a murine prostate cancer model (Table 2), prompting angiogenesis and tumor growth [87]. Secretion of proangiogenic factors and M2 macrophage recolonization was observed also in an oral cancer model [100]. The acceleration of tumor growth after high doses of irradiation was also observed in murine pancreatic tumor models, high M2-like TAM infiltration was revealed associated with a T cell suppressive response. Pancreatic dysplasia rapidly progressed toward invasive pancreatic ductal carcinoma. TAMs were characterized by decrease of IRF5, iNOS and H2eb1 mRNA expression and increase of CD206, Arg-1 and PD-L1 [104]. These findings suggest that high-dose-irradiated TAMs promote tumor progression and tumor radio-resistance. It might be of interest to investigate the possibility to prevent macrophage recruitment by blocking M-CSF to avoid tumor promotion and maintain the efficacy of radiotherapy.

### 5.3. TAMs and Immune Checkpoint Blocking (ICB) Therapy

The T cell surface presents a family of proteins, the immune checkpoints that interact with specific ligands on antigen presenting cells or tumor cells and block T cell receptor-mediated activation. In recent years, the use of immune checkpoint inhibitors provided interesting clinical responses in some type of cancer. TAMs are known to limit the efficacy of ICB therapy [105,106] because they express ligands of checkpoint receptors like PD-L1, PD-L2, CD80, CD86, the V-domain immunoglobulin suppressor of T cell activation (VISTA). These ligands are able to sequester immune checkpoint inhibitors (i.e. anti-PD-L1 monoclonal antibody). TAMs can also capture anti-PD-1 successfully competing with T cells that can bind anti-PD-1 only for a short time [105]. T cells, mast cells and basophils express the ligand of the CD40 receptor (CD40L). CD40 is the receptor of the TNF receptor superfamily and is expressed on the surface of macrophages, DCs. The interaction CD40L-CD40 receptor upregulates the expression of MHC molecules and the production of proinflammatory cytokines that promote T cell activation [107]. In murine tumor models, the use of anti-CD40 antibodies prompted the recovery of tumor immune surveillance via TAM repolarization towards the M1 phenotype favoring an enhanced antitumor activity [108]. Checkpoint immunotherapy in combination with anti-CD40 monoclonal antibody represents a novel alternative that is under investigation. Indeed, the combination of anti-CD40 antibodies with anti CSF1R antibodies has been demonstrated to modify a “cold” tumor in a “hot” tumor enhancing T cell infiltration and the antitumor activity [109].

### 5.4. TAMs and Virotherapy

The use of virotherapy is emerging as a valid therapeutic treatment for several type of cancer. It is based on the use of OVs that, beyond their direct lytic effect on tumor cells, function through a multitude of events like modification of the tumor micro/macro-environment, modulation of the antitumor immune response [6,7,8,9]. The role of macrophages on virotherapy efficacy varies according to the tumor model. M1 macrophages are often considered allies of OV. Despite their potential for increasing virus clearance, they favor tumor shrinkage. In contrast, M2 macrophages seem to be foes, they may promote cancer growth that prevails over any effect on preventing immune clearance of the OV. We report TAM modulation in virotherapy applied to colorectal cancer, glioblastoma, pancreatic and breast cancers (Table 3).

In colorectal cancers treated with virotherapy, the activation of inflammatory macrophage is beneficial. In this tumor model, the oncolytic poxvirus vvDD-CCL11 increased immunogenic programmed necrosis; the antitumor acquired immune response correlated with enhanced levels of IFNγ following virus infection [110]. The oncolytic vaccinia virus, GLV-1h68, was associated with enhanced infiltration of NKs and macrophages and increased levels of proinflammatory cytokines and chemokines (Table 3) [111] involved in antiviral and antitumor immune response. Additionally, inflammatory macrophages express macrophage metallelastase endowed with anti-angiogenic activity and able to improve oncolytic adenovirus spread in colorectal cancers [125]. 

In contrast, in glioblastoma multiforme, virotherapy is improved with the suppression of the innate immune response. The antitumor efficacy of oncolytic herpes simplex virus (oHSV) in glioma was inhibited by inflammatory macrophage activation partly because of the TNFβ mediated arrest of virus replication [112,113]. Indeed, cysteine-rich 61 protein (CCN1) activation inhibited OV antitumor effects via the activation and infiltration of TAMs and NKs expressing CXCL10, MCP-1/3, IFNγ, IL1β [114,115]. In glioblastoma, the activation of M2 macrophages with TGFβ inhibited innate inflammatory macrophages, NKs and microglia, prompting an enhanced virus replication and oHSV antitumor activity [116]. On the other hand, the use of TNFα inhibitors may significantly improve the efficacy of oHSV in glioblastoma. Virus treatment increases macrophage infiltration polarized toward a M1, proinflammatory phenotype, indeed, macrophages/microglia secreted significant levels of TNFα in response to infected glioma cells in vitro and in vivo [113].

In the Netherlands, the oncolytic virus DNX-2401 (formerly Delta-24-RGD), a replication-competent adenovirus modified to increase tropism to glioma cells and replicate in tumor cells that have a defective Rb pathway, entered a phase I/II clinical trial for recurrent glioblastoma (NCT01582516) [117]. In some patients, the virus enhanced the cerebrospinal fluid concentrations of cytokines such as IFNγ, TNF, IL6 and increased levels of CD64, a marker of M1 polarization was observed on macrophages in vitro. 

Another study used the rodent H-1 parvovirus (H-1PV), which is the smallest among all OVs, is endowed with natural anticancer activity and is nonpathogenic for humans. The lack of pre-existing immunity in humans and its capacity to cross the blood–brain barrier, made this virus suitable for central nervous system tumors. In glioblastoma patients, H-1PV showed the switch of an immunosuppressed tumor microenvironment towards immunogenicity. The tumor was infiltrated with CTLs, indeed, induction of cathepsin B and iNOS expression (marker of M1 phenotype) in TAMs and accumulation of activated TAMs in CD40L-positive glioblastoma regions was observed [118]. Combination of virotherapy with ICB therapy was investigated in glioblastoma. In particular, the triple combination of anti-CTLA-4, anti-PD-1, and oHSV G47Δ expressing murine IL12 (G47Δ-mIL12) cured most mice in two glioma models. This treatment was associated with influx of macrophages and M1-like polarization [119].

In pancreatic ductal adenocarcinoma, the release of IFNγ seems to mediate the anticancer effect of H-1PV. In rats, the injection of H-1PV with coapplication of IFNγ extended animal survival and improved the H-1PV induced peritoneal macrophage antitumor response. The authors also suggest that IFNγ may induce antigen presentation by enhancing MHC-II molecule expression on the surface of macrophages and DCs [120]. In the same tumor model, the oncolytic adenovirus expressing TNFα and IL2 (OAd-TNFa-IL2) used in combination with mesothelin-redirected chimeric antigen receptor T cell (meso-CAR T cell) therapy induced significant tumor regression in mice engrafted with highly aggressive and immunosuppressive pancreatic ductal adenocarcinoma. This approach increased CAR T cell and host T cell infiltration to the tumor and polarized macrophages toward the M1 phenotype and increased DC maturation [121]. 

In pancreatic cancer, a combination of immunotherapy based on CD40 as target and oncolytic adenovirus showed antitumor efficacy. A novel oncolytic adenovirus, TMZ-CD40L, armed with a trimerized membrane-bound extracellular CD40L was able to control tumor progression and, indeed, potently increased tumor-infiltrating T cells and promoted the switch from M2 (the classical phenotype of pancreatic tumors) to M1 macrophages [122].

In a model of anaplastic thyroid carcinoma the oncolytic adenovirus dl922-947 decreased monocyte chemotaxis in vitro and tumor macrophage density in vivo and induced the switch of TAMs toward the M1 phenotype, likely by increasing IFNγ [126].

In breast cancer, the presence of TAMs and immune cells is often correlated with poor prognosis, these cells increase tumor resistance to chemotherapy and elicit immunosuppression. Tumor infiltrating immune cells constitutively express low levels of IFN inducing antiviral activity. In particular, breast cancers have constitutive activation of IFN–stimulated genes, activation of antiviral JAK/STAT signaling and are heavily infiltrated with CD68+ macrophages. The use of JAK inhibitors was able to reverse the macrophage-induced antiviral status [127]. The antitumor efficacy of the oncolytic paramyxoviruses’ (measles/mumps) was increased by human monocyte-derived macrophages independently of the initial polarization status of macrophages and viral replication [123]. Furthermore, an oncolytic adenovirus expressing soluble TGFβ receptor II –Fc fused, inhibited TGFβ in bone metastasis of breast cancer reducing M2 osteoclast activity and tumor progression [124].

## 6. Conclusions

Therapeutic strategies targeting TAMs have the advantage of hitting simultaneously all the processes in which they are negatively involved in the tumor microenvironment beyond tumor growth and progression, which are immune regulation and mechanisms of resistance to chemotherapy. Several key enzymes involved in processes of tumor progression are suitably druggable, and several molecules are already available. However, a better depiction of the mechanisms underlying the relationship between TAMs and cancer and the peculiarities of this dangerous liaison in different tumor types, will help to implement effective targeted strategies on immunosuppressive M2 subpopulations, in order to avoid the undesired side effects due to depletion of all macrophages. Undoubtedly, this approach can be optimized through appropriate combination regimens with chemotherapy, immunotherapy, radiotherapy, ICB therapy and virotherapy designed to drive an antitumor TAM profile and tailored to patient characteristics. Thus, optimization of TAM-targeted therapies and combination with current antineoplastic therapies might be addressed to avoid the overcome of adaptive or intrinsic therapy resistance due to TAM plasticity and improve the efficacy of antineoplastic treatments.

## Figures and Tables

**Figure 1 cancers-12-01987-f001:**
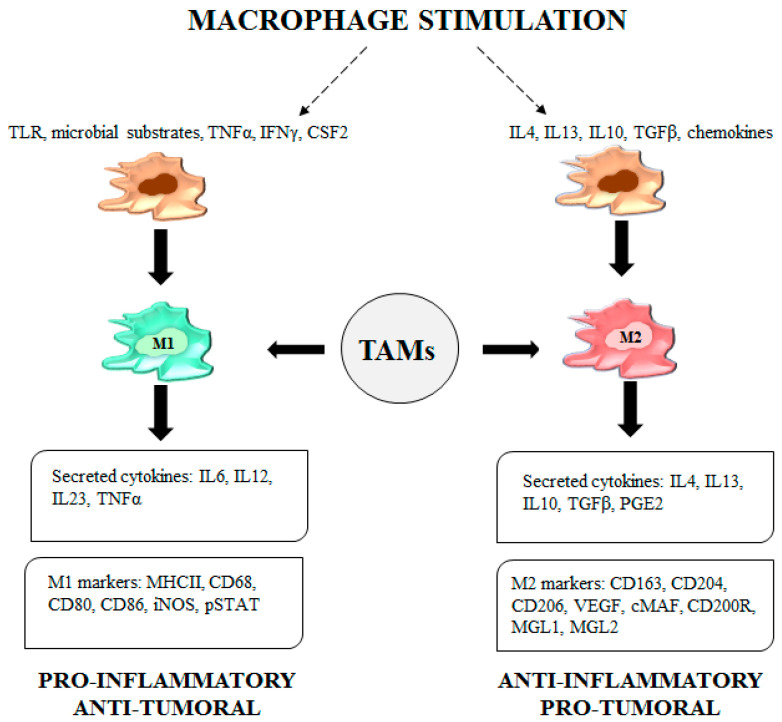
Legend. Tumor-associated macrophage (TAM) functions. Upon stimulation by a TLR, microbial substrates, IFNγ and CSF2, M1-TAMs secrete cytokines such as IL6, IL12, IL23 and TNFα and express specific M1 markers like MHCII molecules, CD68, CD80, CD86, iNOS and pSTAT. The adopted phenotype confers proinflammatory/antitumor features to TAMs. Upon stimulation by IL4, IL13, IL10, TGFβ and chemokines, M2-TAMs secrete IL4, IL13, IL10, TGFβ and PGE2. M2- TAMs express CD163, CD204, CD206, VEGF, cMAF, CD200R, macrophage galactose-type lectin (MGL) 1 and MGL2, all markers representative of the M2 phenotype characterized by anti-inflammatory/protumor functions.

**Figure 2 cancers-12-01987-f002:**
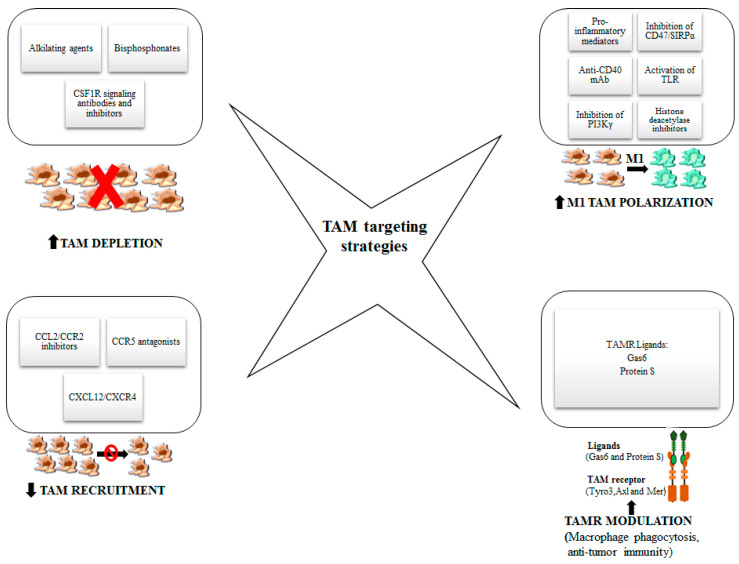
Legend. Strategies to target TAMs and modulate their reprogramming. In this figure, different approaches to target TAMs are represented with the aim of enhancing antitumor efficacy: the depletion of macrophages, the reduction of monocyte recruitment, the polarization into M1-like macrophages and TAM receptor modulation.

**Table 1 cancers-12-01987-t001:** Effects of Chemotherapeutic Agents on Monocytes/Macrophages.

Chemotherapeutic Agent/Combination Therapies	Tumor Type	Effect on Macrophages	Effect on Cancer	References
Paclitaxel	Breast cancer and melanoma	Induction of M1 polarization	Antitumor effect	
	Solid tumors	Activation of proinflammatory molecules (TNFα, IL12, COX2, iNOS, CSFs) and NKs, DC, CTLs	Suppression of tumor growth, increase of antitumor immune response	[67]
Paclitaxel and Albumin (Abraxane)	Murine breast tumor	High infiltration of CD45+ CD169+ macrophages and increase of F4/80+ macrophages	Increase of macrophage infiltration following chemotherapy-induced apoptosis	[68]
Docetaxel	Murine mammary tumor	Activation of M1-like TAMs, depletion of M2-like TAMs increase of CTL response	Inhibition of tumor growth in 4T1-Neu tumor-bearing mice	[69]
Cisplatin and carbonplatin	Human cervical and ovarian cancer	Switch monocyte differentiation toward the M2-like phenotype	Chemoresistance	[70]
Cyclophosphamide	Patients bearing end-stage multi-treated tumors	Reduction of circulating regulatory T cells, restoration of peripheral T cell proliferation and innate killing activities		[71]
	Murine melanoma	Increase of the recruitment of DCs, macrophages and NKs	Favorable impact on cell populations involved in tumor rejection	[72]
Doxorubicin		Increase of the CD206+ macrophages.	Increase of vascular leakageTumor	
	Murine breast cancer	CCR2-dependent monocyte recruitment	Tumor relapse	[73]
Doxorubicin and Lapatinib	HER2/Neu-driven mammary carcinoma	Favored immature macrophage infiltration and reduction of mature TAMs	Reduced cancer growth	[74]
Gemcitabine	Pancreatic cancer	Increase of CD14+ monocytes, CD123+ plasmacytoid DCs and CD11c+ myeloid DCs	Antitumor immune response	[75]
Trabectedin	Murine fibrosarcoma, ovarian carcinoma, Lewis lung carcinoma	Reduction of CD45+ CD11b+ CD115+ monocytes in bloodstream, mature CD11b+ CD115+ monocytes in the bone marrow and F4/80+ macrophages in the spleen	Delayed tumor growth and metastasis, decreased percentage of TAMs	
	Soft tissue sarcoma patients	Strong reduction of TAM density	Strong decrease of blood vessel	[22]
Chemotherapy (cyclophosphamide, doxorubicin, vincristine) and immunotherapy (anti-CD40+ cytosine-phosphate-guanosine-containing oligodeoxynucleotide 1826)	Melanoma, neuroblastoma	M1 polarized TAMs Upregulation of IFNγ, TNFα, IL12, MHC II, CD40, CD80 and CD86 (M1-assiciated molecules). Downregulation of IL4Rα, IL4, IL10 and B7-H1 (M2-associated molecules).	Multidrug chemotherapy synergize with macrophages-activating immunotherapy via TAM repolarization and induction of macrophage-mediated antitumor effects.	[76]
Oxaliplatin, docetaxel, irinotecan with folic acid and 5-fluorouracile+anti-CCL2	Pancreatic and prostate cancer	Monocyte-dependent antitumor response depends on the chemoterapeutic drug	Increase of antitumor response	[77,78]

The table reports the effect of the main chemotherapeutic agents used alone or in combination with other therapies on monocytes, macrophages and TAMs. The effect is described for specific types of tumor.

**Table 2 cancers-12-01987-t002:** Macrophage Reprogramming after Low, Intermediate and High Dose of Irradiation.

Dose	Effect on Polarization	Effects In Vitro	Effects In Vivo
LOW < 1Gy	M2 POLARIZATION	Decrease of iNOS level and NO production in Raw264.7 cells [93]	
		Decrease of p38 phosphorylation and TNFα production in LPS stimulated Raw264.7 cells [94]	M1 phenotype with increase of IL1β, IL12 and TNFα in whole body irradiated BALB/c radiosensitive mice.
		Decrease of IL1β n LPS stimulated THP-1 macrophages [95,96]	M2 phenotype in C57BL/6 radio-resistant mice (118).
		Increase of TGFβ (doses < 2Gy) [97]	
INTERMEDIATE1 to 10Gy	M1 POLARIZATION	Upregulation of proinflammatory markers (HLA-DR, CD86)	
		Downregulation of anti-inflammatory markers (mRNA expression of CD163, MRC1, CD206, versican and IL10) in human un-polarized macrophages	Increase of M1 and decrease of M2 markers [89], increase of iNOS and NO production [88] in whole body irradiate mice.
		In LPS-IFNγ-stimulated macrophages (M1 profile induced): potentiation of acquired M1 profile with induction of HLA-DR.	No changes in M1 and M2 markers with local irradiation. Increase of M1 (IFNγ, IL12p40) and decrease of M2 markers (IL10) with local irradiation + CTL transfer [89].
		In M-CSF and IL10-stimulated macrophages (M2 profile induced): no address of the expression of pro or anti-inflammatory markers [98]	A shift toward M1 phenotype (increase of iNOS, TNF-α, IL-12(p70), pSTAT3) and decrease of M2 markers (CD206, Fizz-1, Arg-1 and Ym-1) with systemic irradiation (2Gy/week for 2 weeks) of RT5 insulinoma bearing mice [89].
HIGH > 10Gy	M2 POLARIZATION	Nf-kB p50 activation, increase of IL10 and reduction of TNFα in Raw264.7 macrophages [99]	M2-like TAM promotion in mice with Panc02 cell xenograft [99] and in oral cancer model [100].
			M2-like TAMs, increase of Arg-1 and COX-2 mRNA expression, decrease of iNOS in murine prostate cancer model [87].

The table reports the effect on macrophage polarization, in vitro and in vivo effects after low, intermediate and high doses of radiotherapy.

**Table 3 cancers-12-01987-t003:** Tumor-associated macrophages in virotherapy.

Tumor Type	OVs	Effects
Colorectal cancer	Poxvirus vvDD-CCL11	increase of immunogenic programmed necrosis antitumor acquired immune response -enhanced levels of IFNγ [110]
	Vaccinia virus GLV-1h68	Infiltration of NK cells and macrophages-increase of proinflammatory cytokines and chemokines (IFN, IL3, IL6, CXCL10, GCP-2, KC/GRO, lymphotactin, M-CSF1, MIP-1, RANTES, MCP-1, MCP-3 and MCP-5) expression of metallelastase by inflammatory macrophages [111]
Glioblastoma	oHSV	oHSV antitumor efficacy is inhibited by: inflammatory macrophage activation in glioma [112,113]-CCN1 activation [114,115]
		oHSV antitumor efficacy is prompted by: M2 macrophage activation with TGF-β [116]
	Virus DNX-2401	increase of the CSF concentration of cytokines (IFNγ, TNF, IL6) and increase of CD64 (M1 polarization marker) [117]
	H-1PV	infiltration of CTLs, induction of cathepsin B and iNOS expression in TAMs [118]
	Virotherapy + ICB (anti-CTLA-4, anti- PD-1 and oHSV G47Δ-mIL-12)	influx of macrophages and M1-like polarization in glioma [119]
Pancreatic ductal adenocarcinoma	H-1PV	coapplication with IFNγ extended animal survival and IFNγ may enhance MHCII molecule expression on the surface of macrophages and DCs [120]
	OAd-TNFα-IL-2 in combinationwith meso-CAR T cell	increase of CAR T cell and host T cell infiltration to the tumor-polarization toward the M1 phenotype -increase of DC maturation [121]
Pancreatic cancer	Immunotherapy + virotherapy Adenovirus TMZ-CD40L	increase of tumor-infiltrating T-cells-switch from M2 to M1 macrophages [122]
Breast cancer	Paramyxoviruses (measles/mumps)	increase of the antitumor efficacy by macrophages independently of initial polarization status and viral replication [123]
	Oncolytic adenovirus expressing soluble TGFβ receptor II-Fc fused	inhibition of TGFβ in bone metastasis reducing M2-osteoclast activity and tumor progression [124]

The table reports the effects of oncolytic viruses (OVs) on specific tumor types.

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
