# Peer review of "Tumor-Associated Macrophage Status in Cancer Treatment"

_cancers, 2020, doi:10.3390/cancers12071987_

Round 1

Reviewer 1 Report

This review is an overview of the effects of targeting TAM in cancer. The authors begun with a summary of the roles of TAM in the tumor microenvironment. Then, they developped all the ways to target TAM leading to their depletion, the inhibition of their recruitment or the modification of their phenotype in an anti-tumor phenotype. Finally, they reviewed the effects of different chemotherapies, radiotherapies and virotherapies on the depletion, recruitment and polarization of TAM. 

This review is a very good overview of the litterature, very complete.

The review is very easy to follow and all the parts are very well constructed and fluid.

Perhaps, the authors could have embellished with one or two synthetic figure.

Author Response

We thank the reviewer for his/her comments and we have added two figures in the revised manuscript.

Reviewer 2 Report

The authors present an expansive survey of therapeutic interventions currently under investigation aimed at altering macrophage function within the tumor microenvironment. This review is timely in its focus on the myeloid compartment, whereas much of the focus in tumor immunology has been on the role of lymphocytes and myeloid derived suppressor cells.  The authors provide a comprehensive introduction of the various approaches which will be helpful in orienting researchers and physicians towards thinking about the role this compartment plays in maintaining tumors and how interfering with them may aid in treatment.  This is particularly important for rational design of approaches aimed at leveraging different agents to achieve maximal responses. The review is well-researched and timely.

Author Response

Minor changes have been made and two figures added according to another reviewer ' request.

Reviewer 3 Report

The Authors have prepared an extensive review titled "Tumor Associated Macrophage status in cancer treatment". there have been various aspects of macrophage therapy discussed in the review and all relevant therapeutic modalities have been touched upon in great detail. 

The review does not need any changes and can be published as is. 

Author Response

Minor changes have been made and two figures added according to another reviewer' request.

Reviewer 4 Report

Anna Maria Malfitano et al provide a nice review of the roles of Tumor-associated macrophages (TAMs), the most abundant innate immune cells in tumors in cancer development, both pro-cancer and anti-cancer functions, like a double-edged sword. More importantly, the reviewed the cancer therapeutics applied in the TAMs status, pro-inflammatory/anti-tumor, or anti-inflammatory/pro-tumor. In addition to the tables presented, the review should provide at least 2 pictures for a much clearer presentation of TAM functions and the related therapeutic strategy. Minor comments:

  1. section 2, Role of TAM in the tumor microenvironment,  the authors should provide more detailed reviews on both clear pro-inflammatory/anti-tumor, or anti-inflammatory/pro-tumor TAMs in two paragraphs.
  2. section 3 Exploiting TAMs as a therapeutic target, the authors should provide some rationale for the strategy, not just the result. brief description of the cross-talk between the TAMs with the tumor microenvironment and cancer cells will be helpful and include a picture here for the clear presentation of data.
  3. Some of the statements/claims should provide the references.

Author Response

We thank the reviewer for his/her comments that we believe have improved our manuscript. We have added two figures in the revised paper.

Minor comments:

  1. 1 line 41: According to the reviewer ‘requests, we have included two additional sub-paragraphs to provide a clear overview of pro-inflammatory/anti-tumor, or anti-inflammatory/pro-tumor TAMs (section 2, 2.1 and 2.2). A related figure (Fig.1 ) has been included.

  1. 3 line 94: We have provide a brief description of the cross-talk between the TAMs with the tumor microenvironment and cancer cells. Additionally, we have briefly described the rationale for the strategies employed to target TAMs. Section 2 is now changed in section 3. A related figure (Fig.2) has been added.

  1. 3 line 94: We have added references to the sentences where required (ref 16 and ref 21).

Round 2

Reviewer 4 Report

The concerns have been addressed.